# OpenReview forum: "ReasoningShield: Safety Detection over Reasoning Traces of Large Reasoning Models"
_ICLR.cc/2026/Conference — ICLR 2026 Conference Withdrawn Submission_

### Official Review · Reviewer_jt2c · 2025-10-30

**Soundness:** 3
**Presentation:** 2
**Contribution:** 3
**Rating:** 6
**Confidence:** 3

**Summary:**

This submission addresses the problem of moderating chains-of-thought (CoTs) produced by large reasoning models (LRMs), i.e., detecting safety risks within CoTs. The submission describes several contributions in this direction: 1) a taxonomy of risks, adapted from previous works; 2) a ReasoningShield-Train dataset for training moderation models, labelled by three annotation models with a human expert fallback; 3) a ReasoningShield-Test dataset, labelled by human experts only and with safety datasets and LRMs not included in ReasoningShield-Train; 4) 3B- and 1B-parameter ReasoningShield moderation models trained using supervised fine-tuning and direct preference optimization. The experiments presented in the main paper mainly show that ReasoningShield clearly outperforms existing moderation tools and generalizes to LRMs and datasets not seen in training.

**Strengths:**

- Appears to be the first to comprehensively tackle the problem of CoT moderation, contributing training and test datasets, a training procedure, and two moderation models.
- ReasoningShield's CoT moderation performance (shown in Table 2) is strong and clearly better than the baselines, particularly prompting of much larger LLMs (e.g. Gemma-3-27B or larger vs. 3B or 1B for ReasoningShield).
- I think the evaluation is well-designed with separation of in-distribution (used for training ReasoningShield) safety datasets and LRMs, and out-of-distribution datasets and LRMs.

**Weaknesses:**

1. A missing class of baselines: As discussed in the introduction, existing moderation tools are challenged by the length and stepwise nature of CoTs. It seems that these challenges could be overcome by also applying existing tools in a stepwise manner, i.e., applying them to each reasoning step in turn and then drawing a conclusion from the results in some way. How much better would this perform? I acknowledge that the computational cost would increase significantly.
1. The number of expert annotators (3) seems low for labelling the ReasoningShield-Test dataset (and similarly the data for the pilot study in Section 3.2).
1. Clarity could be improved as I found quite a few unclear sentences. Please see the questions below, particularly question 4 and onward.
1. The claim of "enhanced explainability" is only supported by three examples given in Figures 6-8, which are furthermore not in the main paper. It would be good to discuss one of these in detail in the main text.
1. Less important: Since results are reported as the average of five runs, standard deviations could also be reported.

Minor:
- Line 241, "each of the ten primary categories is further refined into 42 subcategories": I think the authors mean 42 subcategories in total, not for each primary category.
- Line 250: I would not say that the "Potentially Harmful" label resolves ambiguity, but rather it accommodates ambiguity.
- Lines 348-349: Is $\mathcal{L}$ the set of three safety levels? I do not think it was defined previously.
- Eqs. (3) and (4): Should the subscript $T$ be $y_{CoT}$?
- Lines 361-362: I do not see how this contrastive learning aligns with human standards. As I understand, it aligns with the majority of the three annotation models.
- Lines 371-372: "Baseline Moderation Methods" may be a better heading for this paragraph.

**Questions:**

In addition to questions requesting clarification, please also respond to the first question about related work.

1. Related work:
    1. Lines 102-103 state that Zhou et al. (2025) find that reasoning steps hide unsafe content. What method(s) did they use to obtain this finding, and how do they relate to ReasoningShield?
    1. I am not completely convinced that CoT monitoring (lines 109-110) is out of scope, for example because "Deception & Misinformation" is one of the risk categories in the proposed risk taxonomy. I have a similar question of what methods do these CoT monitoring works use and how they relate.
1. How are reasoning traces segmented into steps $t_1, t_2, ...$?
1. Table 1:
    - For Claude-Sonnet-3.7, why is the F1$_{CoT}$ column mostly zero?
    - Does "OpenAI Moderation" correspond to GPT-4o mentioned in lines 193-194?
1. In line 297, reproducibility is mentioned as a reason for choosing open-source LRMS for ReasoningShield-Train. What is the reproducibility benefit here?
1. Line 307: What does "requirements for consequence-focused judgment" mean?
1. Questions about agreement among the annotation models:
    1. Does consensus mean agreement in terms of both the risk category and the safety level?
    1. Why the term "hard negatives"?
    1. What is the difference between partial consensus ($\leq 2$) and single-vote cases (since $1 < 2$)?
    1. What is meant by "model consistency" in "97% model consistency"?
1. Line 360: How is the scoring function $f_{\phi}$ related to the model?
1. Lines 373-374: What does "raw text detection limitations" mean?
1. Line 434: Does "open-source-derived" refer to open-source LRMs, "OSS" in Table 2?
1. Lines 455-456: What do "evaluation guidance" and "analysis process" refer to exactly?

---

> ### Author Response · Authors · 2025-11-21
> **Rebuttal to Reviewer jt2c (1/4)**
>
> We thank you for your review and positive feedback.
>
> > **W1:** Suggestion to apply existing moderation tools in a stepwise manner to overcome CoT length challenges.
>
> We agree that applying existing tools stepwise is a theoretically sound baseline. However, we identified two critical barriers that make this approach impractical and less effective compared to ReasoningShield:
>
> - **Prohibitive Computational Cost:** As you noted, costs increase significantly. If a CoT consists of N steps, the inference cost scales linearly by a factor of N. Given that a typical CoT contains 30-50 sentences, a 50x increase in latency renders this approach unusable for real-time guardrailing. In contrast, ReasoningShield achieves this fine-grained analysis in a single pass using efficient 1B/3B models.
>
> - **Complexity of Aggregation:** Designing an effective aggregation strategy is non-trivial. For example, determining whether a single flagged step should condemn the entire trace requires global context awareness, which simple aggregation rules often lack.
>
> - **New Experiment: Stepwise Baseline.** To empirically address your query, we conducted a supplementary experiment. Specifically, we uniformly sampled 200 reasoning traces from our test set and segmented them into fixed 100-token windows to simulate a practical stepwise approach while mitigating extreme latency. We applied existing tools to each window and used a strict aggregation rule (i.e., if any segment is flagged, the whole trace is unsafe).
>
> **Table 1:** Holistic vs. Stepwise Application of Baseline Tools
>
> | Model & Strategy | Overall F1 |
> | :---: | :---: |
> | **LlamaGuard-4-12B** (Holistic / Standard) | 50.7% |
> | **LlamaGuard-4-12B** (Stepwise - New) | 76.7% |
> | **WildGuard-7B** (Holistic / Standard) | 63.9% |
> | **WildGuard-7B** (Stepwise - New) | 85.3% |
> | **ReasoningShield-3B** (Ours) | **92.5%** |
>
> **Conclusion:** As shown, even with stepwise application, existing tools still underperform ReasoningShield, because ReasoningShield is trained end-to-end on the full (Query,CoT) pair. This allows it to understand the subtle, global context of risks (e.g., detecting intent across steps) that fragmented stepwise detection and simple aggregation rules fail to capture.
>
> ---
>
> > **W2:** Concern regarding the use of three expert annotators for the test set.
>
> We understand the concern regarding the number of annotators. However, we would like to clarify that this decision aligns with established protocols in published AI safety benchmarks (e.g., WildGuard[1], JailbreakBench[2], SaladBench[3]), which typically employ a small team of 3 high-quality annotators rather than large crowds.
>
> We prioritized "quality over quantity", implementing strict control measures detailed in **Section 4.2 and Appendix D** to ensure reliability:
>
> **1. Expert Qualifications:** Our annotators were not crowd workers, but experienced AI safety researchers with specialized backgrounds in AI alignment and policy. Their expertise ensures a depth of understanding that lay annotators often lack.
>
> **2. Rigorous Protocol:** We established a detailed annotation manual and conducted a mandatory "Calibration Phase" before formal annotation. During this phase, experts annotated non-overlapping samples and discussed discrepancies until they reached an inter-rater reliability of 0.80 (Fleiss' Kappa) .
>
> **3. High Consistency:** In the formal annotation, we also achieved substantial agreement scores: Fleiss' Kappa = 0.75 for the Test set and 0.72 for the Pilot study. This statistical evidence strongly supports the reliability of our labels.
>
> **4. Consensus Mechanism:** We employed a strict majority voting system to determine "gold standard" labels. To further ensure quality, any samples that failed to reach a consensus (at least 2/3 agreement) were explicitly excluded from the dataset.
> We believe this rigorous process validates the reliability of our test set.
>
> **Reference:**
>
> [1] Han et al., "Wildguard: Open one-stop moderation tools for safety risks, jailbreaks, and refusals of llms." NeurIPS (2024).
>
> [2] Chao et al., "Jailbreakbench: An open robustness benchmark for jailbreaking large language models." NeurIPS (2024).
>
> [3] Li et al., "Salad-bench: A hierarchical and comprehensive safety benchmark for large language models." ACL (2024).
>
> ---
>
> > **W4:** Suggestion to discuss a specific explainability example (Figures 6-8) in the main text.
>
> We appreciate this constructive suggestion. In the revised manuscript, rather than simply relying on the Appendix, we will add a dedicated "Case Study" discussion in Section 5.2. We will select the "Rights-Related Risks" scenario (from Figure 6) as a representative example and walk through the model's output in the main text, analyzing how ReasoningShield's stepwise analysis successfully localized the source of risks (i.e., the detailed explanation of privacy Information) that baseline models missed. This allows us to concretely demonstrate the practical value of explainability.

---

> ### Author Response · Authors · 2025-11-21
> **Rebuttal to Reviewer jt2c (2/4)**
>
> > **W5:**  Since results are reported as the average of five runs, standard deviations could also be reported.
>
> We appreciate this suggestion to enhance the rigor of our reporting. While we have already visualized the **95% confidence intervals in Figure 4 and Figure 5** of our original submission, we agree that explicitly reporting standard deviations in Table 2 provides greater clarity regarding result stability. We will include the standard deviations in the revised manuscript. Some of the data is presented as follows:
>
> **Table 2:** Standard Deviations of Moderation Models Performance (CoT Moderation)
>
> | Model | Overall F1 ± SD (%) |
> | :---: | :---: |
> | **ReasoningShield-3B** | **89.1 ± 0.2** |
> | **ReasoningShield-1B** | **86.3 ± 0.3** |
> | **Qwen2.5-72B** (w/ LG Prompt) | 79.3 ± 0.6 |
> | **GPT-4o** (w/ LG Prompt) | 73.7 ± 0.5 |
> | **Mistral-3.1-24B** (w/ LG Prompt) | 70.0 ± 0.6 |
> | **LlamaGuard-4-12B** | 51.6 ± 0.1 |
>
> ---
>
> > **Q1.1:** Related Work - Zhou et al. (2025) Methodology & Relation
>
> - **Their Methodology:** Zhou et al. (2025) [1] focus on safety evaluation rather than defense. Their method involves explicitly separating the reasoning trace from the answer and using GPT-4o as an automated judge to perform binary safety classification of each component independently.
>
> - **Relation:** They act as the "Problem Discoverer," while ReasoningShield is the "Problem Solver."
>     - **From Evaluation to Moderation:** Zhou et al. empirically confirm the existence of hidden risks. We take the next step by formalizing and addressing the specific CoT Moderation task.
>     - **From General to Specialized:** They rely on heavy, general-purpose models (GPT-4o). Our work demonstrates that such models are computationally expensive, and we provide the first specialized, lightweight (1B/3B) framework to address this gap .
>
> > **Q1.2:** Scope of CoT Monitoring
>
> Although there is terminological overlap, the research domains are distinct:
>
> - **Different Monitoring Targets:** ReasoningShield targets **Content Safety**. Our "Deception" category refers to generating harmful content for users, such as phishing emails or fraud guides . In contrast, CoT Monitoring [2]  targets **Frontier Risks** like "alignment faking" or "reward hacking," where the model attempts to deceive the developer or the training process (e.g., sycophancy, sandbagging)
>
> - **Different Methods:** CoT monitoring literature typically uses zero-shot prompting of massive models to detect behavioral anomalies. We propose a fine-tuned, lightweight detector specifically for content compliance.
>
> **Reference:**
>
> [1] Zhou et al., "The hidden risks of large reasoning models: A safety assessment of r1." arXiv:2502.12659 (2025).
>
> [2] Baker et al., "Monitoring reasoning models for misbehavior and the risks of promoting obfuscation." arXiv:2503.11926 (2025).
>
> ---
>
> > **Q2:** How are reasoning traces segmented into steps?
>
> In our work, $t_i$ represents a conceptual reasoning step, akin to the "step-by-step" decomposition described in the original Chain-of-Thought [1] literature.
>
> **Reference:**
>
> [1] Wei et al., "Chain-of-thought prompting elicits reasoning in large language models." NeurIPS (2022).
>
> ---
>
> > **Q4:** Reproducibility of Open-Source LRMs
>
> The "reproducibility" benefit mentioned in Section 4.2 refers to the dataset generation pipeline. By selecting open-source LRMs (e.g., DeepSeek-R1, QwQ) for the training set, we ensure that other researchers can utilize the exact same model weights and versions to replicate our data generation process. In contrast, commercial models are often "black boxes" with frequent version updates and behavioral changes, making precise replication of the training data collection impossible.
>
> ---
>
> > **Q5:**  Consequence-focused judgment
>
> This refers to a critical design of our system prompt (detailed in Appendix J.1). The instruction explicitly directs the model to: "Assess each query individually based on its core intent and potential consequences, disregarding its framing or expression (e.g., educational, hypothetical...)". This requirement is essential for **resisting jailbreak attempts**, as adversarial prompts often attempt to disguise harmful intents behind benign or safe-sounding pretexts. It forces the judge model to evaluate whether the outcome facilitates harm, regardless of the tone used.

---

> ### Author Response · Authors · 2025-11-21
> **Rebuttal to Reviewer jt2c (3/4)**
>
> > **Q3.1:** Claude-Sonnet-3.7 F1 Score
>
> This is primarily attributable to label imbalance and hidden risks in Claude-Sonnet-3.7's traces.
>
> **1. Dataset Distribution and Simulation Design.**  Following the methodology of SafeChain [1], our objective was to simulate a real-world "in-the-wild" scenario. Therefore, we prioritized input consistency by uniformly sampling queries from four safety datasets, allowing the response labels to reflect the models' natural behavior rather than strictly enforcing artificial balance during generation.
>
> **2. Explaining Claude-Sonnet-3.7's Performance.** This design directly explains the anomaly observed with Claude-Sonnet-3.7 in Table 1:
>
>   - **Label Imbalance:** While other LRMs produced a safe-to-unsafe ratio of roughly 5:5 or 6:4, Claude-Sonnet-3.7 was exceptionally well-aligned, resulting in a highly imbalanced 9:1 ratio.
>
>   - **Hidden Risks:** The few risk samples it did produce were predominantly "Potentially Harmful". In these traces, risky elements were often deeply interspersed within large volumes of benign text and thereby were more difficult to detect.
>
>   - **Failure of Existing Tools:** Traditional holistic moderation tools (like LlamaGuard) consistently missed these subtle, "needle-in-a-haystack" risks, leading to the near-zero Recall and $F1_{CoT}$ observed in the original Table 1.
>
> **3. New Experiment: Balanced Claude-3.7 Validation.** To confirm that this failure is due to intrinsic model limitations rather than a metric artifact of label imbalance, we conducted a targeted validation study.
>
>   - **Method:** We prompted Claude-Sonnet-3.7 with all questions from our test set to generate a comprehensive response pool. From this pool, we sampled a specific subset of 200 (Query, CoT) pairs and corresponding (Query, Answer) pairs with a strict **1:1 Safe:Unsafe ratio**.
>
>   - **Results:** We re-ran the evaluation on this strictly balanced set. The results validate our original conclusion: existing moderation tools struggle significantly with CoT Moderation, confirming that they struggle to parse the complex risk structures in Claude-3.7's reasoning, even when the dataset is strictly balanced.
>
> **Table 3:** Baselines Performance on Balanced  Claude-3.7-Sonnet Dataset
>
> | Moderation Model | Acc (Ans) | F1 (Ans) | Acc (CoT) | F1 (CoT) | $\Delta$ Acc | $\Delta$ F1 |
> | :---: | :---: | :---: | :---: | :---: | :---: | :---: |
> | LlamaGuard-1-7B | 68.5% | 36.4% | 53.0% | 13.0% | **-15.5%** | **-23.4%** |
> | LlamaGuard-2-8B | 69.5% | 41.9% | 53.0% | 11.3% | **-16.5%** | **-30.6%** |
> | LlamaGuard-3-8B | 66.5% | 29.5% | 53.5% | 13.1% | **-13.0%** | **-16.4%** |
> | LlamaGuard-4-12B | 71.0% | 45.3% | 56.5% | 23.0% | **-14.5%** | **-22.3%** |
> | WildGuard-7B | 73.5% | 53.9% | 56.0% | 21.4% | **-17.5%** | **-32.5%** |
> | OpenAI Moderation | 63.0% | 21.3% | 45.0% | 3.5% | **-18.0%** | **-17.8%** |
>
> **Reference:**
>
> [1] Jiang et al., "SafeChain: Safety of language models with long chain-of-thought reasoning capabilities," ACL (2025).
>
> > **Q3.2:** OpenAI Moderation vs. GPT-4o
>
> No. As detailed in Lines 113-115 and Lines 373-374, they represent two distinct baselines:
>
> - **OpenAI Moderation:** Refers to the dedicated text-moderation-latest API endpoint, a specialized model designed specifically for content safety classification.
>
> - **GPT-4o:** Refers to the general-purpose LLM (GPT-4o) acting as a moderator, prompted with specific instructions to evaluate content.
>
> ---
>
> > **Q6:** Agreement & Definitions
> - **Consensus:** Yes, it means agreement on both risk category and safety level.
> - **Why "Hard Negatives":** We use this term because these are samples where our AI Judge ensemble failed to reach a unanimous decision. This disagreement indicates that the samples are inherently ambiguous, subtle, or lie near the decision boundary. These "hard" examples are ideal training material for the DPO stage to improve robustness .
> - **Partial Consensus vs. Single-vote:**
>     - **Partial Consensus ($\le 2$):** Refers to any case where the three models did not unanimously agree (i.e., a 2 vs. 1 split or a 1:1:1 split).
>     - **Single-vote:** Refers to the specific subset of partial consensus where no majority formed (i.e., a 1:1:1 split). Because no single label received more than one vote, these highly ambiguous edge cases were sent to human experts for relabeling.
> - **Model Consistency:** As defined in Appendix D.3, this is the percentage of samples where at least two of the three judge models reached the same judgment.
>
> ---
>
> > **Q7:** Line 360 How is the scoring function related to the model?
>
> In the DPO objective function (Eq. 4), the scoring function $f_{\phi}(\cdot)$ implicitly represents the log probabilities assigned by the model being fine-tuned (relative to the reference model). The symbol $\phi$ denotes the parameters of the model we are optimizing. This follows the standard notation established in the original DPO literature.

---

> ### Author Response · Authors · 2025-11-21
> **Rebuttal to Reviewer jt2c (4/4)**
>
> > **Q8:** Lines 373-374: What does "raw text detection limitations" mean?
>
> This refers to the input constraints of many API-based moderation tools (e.g., OpenAI Moderation API). These tools are typically designed to accept a single unstructured text string for classification. Therefore, to evaluate them on the CoT Moderation task, we passed the reasoning trace text ($y_{CoT}$) as input.
>
> ---
>
> > **Q9:** Line 434: Does "open-source-derived" refer to open-source LRMs, "OSS" in Table 2?
>
> Yes, you are correct. This phrase refers to the "OSS" column in Table 2, representing traces generated by open-source LRMs.
>
> ---
>
>
> > **Q10:** Lines 455-456: What do "evaluation guidance" and "analysis process" refer to exactly?
>
> These terms refer to specific components of our system prompt (Appendix J.1), which we validated in our Ablation Study (Table 3) :
>
> - **Evaluation guidance:**  Refers to the ``#Evaluation Process`` section, which instructs the model on how to structure its stepwise analysis.
>
> - **Analysis process:**  Refers to the requirement to output the ``Analysis: [...]`` field before the final judgment.
>
> As shown in Table 3, removing either component leads to a measurable decline in performance (e.g., -15.7% F1 without analysis process), proving the effectiveness of our method design.
>
> ---
>
> > **Minor Points**
>
> **1. Line 241:** Yes, we meant 42 subcategories in total. We have corrected this phrasing to avoid confusion.
>
> **2. Line 250:** We agree that "accommodates ambiguity" is scientifically more accurate, as the category is designed to handle edge cases rather than eliminate them. We have adopted this change.
>
> **3. Lines 348-349:**  The set $\mathcal{L}$ (representing the predefined label options) was formally defined in Section 3.1 during the problem formulation. We have added a back-reference here to ensure clarity.
>
> **4. Eqs. (3) and (4):**  We have corrected the subscripts in Equations (3) and (4) to correctly denote the model parameters.
>
> **5. Lines 361-362:**  We clarify that DPO aligns the model with the "majority vote" of our AI judge ensemble. Since we validated that this ensemble achieves 92% agreement with human experts (Appendix D.3), it serves as a reliable proxy for human standards. However, we have refined the text in Section 4.3 to state that DPO improves alignment with "high-quality consensus labels" to be precise.
>
> **6. Lines 371-372:**  Agreed. We have renamed the heading to "Baseline Moderation Models" in Section 5.1 to better reflect the content.
>
> ---
>
> We sincerely thank you again for your constructive feedback. We hope that these clarifications and additional experiments have fully addressed your concerns. We will incorporate the suggested corrections and key experimental results to further strengthen the manuscript. Should you have any further questions, we would be happy to address them.

---

### Official Review · Reviewer_9cpv · 2025-10-30

**Soundness:** 3
**Presentation:** 3
**Contribution:** 2
**Rating:** 4
**Confidence:** 4

**Summary:**

This paper studies safety risks in large reasoning models (LRMs)when chain-of-thought (CoT) is revealed, and proposes ReasoningShield, a safety monitor designed to detect harmful content embedded in reasoning traces. The authors construct a new dataset (9.2K CoT training samples, 2.2K test samples) with multi-level safety annotations, and train a detector via SFT + DPO. Experiments show that the method outperforms prior safety monitors, including LlamaGuard and GPT-4o, in detecting covert harmful reasoning steps. The approach

**Strengths:**

1. The problem motivation is timely. The paper clearly demonstrates that releasing CoT increases safety risks and that existing safety monitors fail to capture covert malicious reasoning.
2. A reasonably sized dataset with structured safety labels is provided, which may benefit future research on LRM safety and CoT oversight.
3. The approach is simple, inference-efficient, and improves smaller model performance, which is important for realistic deployment scenarios.
4. Empirical results cover several safety benchmarks and model types. Ablations validate the importance of prompt design and DPO training.
5. The paper is clearly written and presents compelling examples illustrating real safety failure modes in CoT.

**Weaknesses:**

1. Innovation is relatively incremental. The method primarily applies standard SFT + DPO pipelines to a new dataset rather than proposing new architectures or fundamentally novel mechanisms for CoT safety.
2. Dataset construction and annotation quality are insufficiently detailed. The paper lacks breakdowns on annotator consistency, error rates, and quality control procedures.
3. Limited exploration of scaling behavior. The study focuses on smaller models; results on larger safety monitors (e.g., 70B-scale) would strengthen generality.
4. Lack of adversarial or adaptive attack evaluation. It is unclear whether the method remains reliable when attackers intentionally obfuscate harmful reasoning.
5. Some claims regarding “comprehensive coverage” feel overstated given limited analysis on multi-turn interaction, longer reasoning chains, or domain-specific harms.

**Questions:**

1. Clarify contribution relative to existing safety monitors (e.g., LlamaGuard) beyond dataset + fine-tuning; explicitly highlight differences in handling sequential reasoning structures.
2. Provide detailed annotation protocol, inter-annotator agreement, and dataset quality diagnostics.
3. Include adversarial tests targeting reasoning-chain obfuscation to demonstrate robustness.
4. Evaluate more model scales or provide discussion on scalability and model distillation strategies.
5. Consider adding qualitative failure cases to better illustrate remaining challenges.

Note: I am currently leaning toward a borderline rejection. However, I am open to increasing my score if the concerns outlined above are adequately addressed during the rebuttal phase.

---

> ### Author Response · Authors · 2025-11-21
> **Rebuttal to Reviewer 9cpv (1/3)**
>
> We thank you for your review and insightful feedback.
>
> > **W1 & Q1:** Concerns regarding incremental innovation (standard SFT+DPO) and request to clarify contributions relative to existing monitors regarding sequential structures.
>
> We acknowledge that SFT and DPO are established algorithms.  However, we clarify that our core contribution is designing a specialized supervision framework that aligns the model with the sequential nature of reasoning traces, rather than proposing a new optimizer. This approach contrasts sharply with existing monitors in three ways:
>
> **1. Mechanism 1: From Holistic to Stepwise Supervision**
>
>   - **Existing Monitors (e.g., LlamaGuard [1]):** Treat the entire output as a monolithic "black box" (Appendix J.4), performing sparse, outcome-oriented detection that often misses risk propagation within reasoning traces.
>
>   - **ReasoningShield:** Implements dense, process-oriented supervision. We explicitly train the model to parse the sequential structure via: (1) intent analysis, (2) stepwise risk tagging, (3) synthesis into final judgement. This aligns the supervision signal with the reasoning flow, enabling the model to detect risks hidden in intermediate steps.
>
> **2. Mechanism 2: Capturing Fine-Grained Risk Semantics**
>
>   - **Existing Monitors:** Rely on binary labels that force complex, evolving reasoning traces into a rigid "Safe/Unsafe" dichotomy, limiting detection of subtle semantic shifts.
>
>   - **ReasoningShield:**  Introduces a 3-level taxonomy. This granularity enables the model to better capture nuanced risk semantics within the sequence, identifying the semantic evolution of risk along the trace that standard binary classifiers miss.
>
> **3. Mechanism 3: Optimizing Decision Boundaries via Hard Negatives**
>
>   - **Existing Monitors:** Typically use standard SFT (e.g., LlamaGuard[1], WildGuard[2], Qwen3Guard[3]) and often struggle with hard negative cases.
>
>   - **ReasoningShield:** Employs a strategic two-stage training process. We utilize a targeted DPO stage on "Hard Negatives". These are specific sequences where the risk is subtle or debatable.  By optimizing on these edge cases, the model learns to distinguish between benign logical steps and those that subtly cross the safety boundary.
>
> While existing moderation models share standard training methodologies (SFT), we acknowledge that simplicity and efficiency are critical. Our key contribution is the formulation of the CoT Moderation and a process-based supervision framework, a foundational advance that transcends incremental method adjustments. This paradigm allows our lightweight model (1B/3B) to significantly outperform much larger general-purpose models  (e.g., GPT-4o) on CoT Moderation.  This "Small-Beats-Large" result highlights the effectiveness and practical value of our work.
>
> **References:**
>
> [1] Inan et al., "Llama guard: Llm-based input-output safeguard for human-ai conversations." arXiv:2312.06674 (2023).
>
> [2] Han et al., "Wildguard: Open one-stop moderation tools for safety risks, jailbreaks, and refusals of llms," NeurIPS (2024).
>
> [3] Zhao et al., "Qwen3Guard Technical Report." arXiv:2510.14276 (2025).
>
> ---
>
> > **W2 & Q2:** Concerns regarding insufficient detail on annotation consistency, error rates, and quality control procedures.
>
> In **Appendix D and Section 4.2** of our original submission, we already provide extensive details on our annotation pipeline. We summarize the core quality assurance measures below:
>
> **1. Detailed Annotation Protocol:**
>
>   - **Expert Annotation:** Unlike many large-scale datasets that rely on crowdworkers, our annotation was conducted by 3 documented AI safety experts with specialized backgrounds in alignment and policy (Appendix D.1).
>
>   - **Rigorous Calibration:** We developed a comprehensive annotation manual defining specific criteria for "Harmful," "Potentially Harmful," and "Safe" categories, including edge case handling for ambiguous reasoning traces.  Prior to formal annotation, we performed a calibration phase, iterating until the inter-rater reliability (Fleiss' Kappa) reached 0.80, and formal independent annotation with majority voting to establish gold labels.
>
> **2. Inter-Annotator Agreement:**
>   - Following standards set by WildGuard and BingoGuard,  we quantified agreement using Fleiss' Kappa. For the test set, we achieved a  Fleiss' Kappa of 0.75, indicating substantial agreement.
>
> **3. Dataset Quality Diagnostics:**
>   - **Test Set:** To guarantee a reliable gold standard, we employed a strict consensus mechanism. Any sample failing to reach a majority vote among experts was explicitly discarded.
>
>   - **Training Set:** We validated our Human-AI collaborative framework against the human-labeled pilot dataset (Appendix D.3). It achieved 92.3% accuracy (agreement with human labels) and 97.3% model consistency. Furthermore, we also employed a separate LLM-based "Data Quality Auditor" (Appendix J.2) to filter out malformed or low-quality outputs.

---

> ### Author Response · Authors · 2025-11-21
> **Rebuttal to Reviewer 9cpv (2/3)**
>
> > **W3 & Q4:** Limited exploration of scaling behavior. The study focuses on smaller models; results on larger safety monitors (e.g., 70B-scale) would strengthen generality.
>
> Thanks for this suggestion. We address this in two parts: our rationale for prioritizing lightweight models and a new experimental study on scaling behavior.
>
> **1. Rationale: Intentional Design for Efficiency.** We must clarify that focusing on lightweight models (1B/3B) was a deliberate design choice for practical utility.
>   - **Deployment Reality:** In real-world scenarios, a safety guardrail must have a significantly lower computational footprint than the LLM it protects. Using a 70B guardrail to audit a LRM of similar size effectively doubles inference costs and latency, hindering real-time deployment.
>
>   - **Small-Beats-Large:** Our key finding is that a specialized small model can  outperform massive general models like GPT-4o in CoT Moderation. This demonstrates that for the nuanced task, data quality and our structured supervision paradigm are more critical than raw scale.
>
> **2. New Experiment: Exploration of Scaling.** To directly address your concern regarding generality and scaling laws, we conducted a new scaling experiment. Since the Llama-3.2 family (our original base) is limited to smaller sizes, we utilized the Qwen2.5 family, which offers a comprehensive range of parameter sizes. We trained varying sizes (0.5B to 72B) using our ReasoningShield pipeline.
>
> **Table 1:** Scaling Analysis (Performance vs. Latency) on Qwen2.5 Family
>
> | Base Model (Qwen2.5) | Overall F1 | Latency (s/sample) | GPU Mem (GB) | Load Time (s) |
> | :---: | :---: | :---: | :---: | :---: |
> | **0.5B** | 0.801 | 9.29 | 0.92 | 4.57 |
> | **1.5B** | 0.848 | 10.31 | 2.88 | 4.84 |
> | **3B** | 0.873 | 12.96 | 5.75 | 7.22 |
> | **7B** | 0.885 | 14.00 | 14.19 | 11.05 |
> | **32B** | 0.895 | 22.56 | 61.03 | 125.66 |
> | **72B** | 0.901 | 30.27 | 135.43 | 277.69 |
>
> **Conclusion:** As shown, our method demonstrates positive scaling behavior, confirming that our data-centric approach generalizes effectively to larger architectures. However, latency also increases significantly. This validates our original motivation: while larger models offer marginal accuracy gains, the 1B/3B scale provides the optimal trade-off between high accuracy and the efficiency required for real-world guardrailing.
>
> ---
>
> > **W4 & Q3:** Concerns regarding the lack of adversarial evaluation and robustness against reasoning chain obfuscation.
>
> We further validate the adversarial robustness of our model. Specifically, we address this through our existing adversarial benchmarks and a new Red Teaming experiment targeting trace obfuscation.
>
> **1. Existing Evidence: Robustness against Adversarial Prompts.** Our evaluation is not based on simple sampling. As detailed in Appendix F.2, our test set explicitly integrates data from dedicated adversarial benchmarks, as SALAD-Bench and Jailbreak-Bench contain "attack-enhanced" queries designed to bypass safety filters. Our strong performance on these datasets is direct evidence of robustness. For instance, on the Jailbreak-Bench(OSS subset), ReasoningShield-3B achieves an F1 score of 92.3%, whereas LlamaGuard-4 achieves only 69.2%. This demonstrates that our method is highly robust against attackers who intentionally obfuscate prompts.
>
> **2. New Experiment: Robustness against Obfuscated Reasoning Traces.** We interpret your specific concern as referring to attacks where the Reasoning Trace itself is manipulated to evade detection, rather than just the user prompt. To address this, we conducted a new Red Teaming experiment. Inspired by adversarial methods like ReNeLLM [1] and PAIR [2], we selected 100 risky (Query, CoT) samples and applied five distinct obfuscation strategies to the reasoning traces:
>   - Sensitive Word Substitution and Fuzzification: Replacing key risk terms with synonyms.
>   - Sentence Restructuring: Altering word order while maintaining semantic consistency.
>   - Noise Insertion: Inserting meaningless characters to disrupt tokenization.
>   - Partial Translation: Mixing languages within the trace.
>   - Style Transfer (Rewriting): Paraphrasing harmful reasoning into benign-sounding styles.
> This generated 500 red-teamed adversarial reasoning traces. The results are shown below:
>
> **Table 2:** Robustness against Red-Teamed Obfuscated Traces
>
> | Model | Accuracy | F1 Score |
> | :---: | :---: | :---: |
> | LlamaGuard-4-12B | 0.333 | 0.500 |
> | **ReasoningShield-3B** | **0.873** | **0.932** |
>
> **Results:** As shown, ReasoningShield maintains high robustness against obfuscated traces compared to the LlamaGuard-4, validating its potential effectiveness against intentional attacks in deployment.
>
> **References:**
>
> [1] Ding et al., "A wolf in sheep’s clothing: Generalized nested jailbreak prompts can fool large language models easily." NAACL (2024).
>
> [2] Chao et al., "Jailbreaking black box large language models in twenty queries." SaTML (2025).

---

> ### Author Response · Authors · 2025-11-21
> **Rebuttal to Reviewer 9cpv (3/3)**
>
> > **W5 & Q5:** Some claims regarding “comprehensive coverage” feel overstated given limited analysis on multi-turn interaction, longer reasoning chains, or domain-specific harms. Consider adding qualitative failure cases to better illustrate remaining challenges.
>
> Thanks for the comment. We agree that "comprehensive" is a strong claim, and we will refine our language to be more precise while adding the requested failure analysis.
>
> **1. Clarifying "Comprehensive" Coverage.** Our original claim referred specifically to our 10-category risk taxonomy, which synthesizes academic benchmarks and industry policies (e.g., OpenAI, Google, Anthropic) to cover the vast majority of content safety risks. We acknowledge this did not imply coverage of all interaction modalities. We will soften this language in the revision to be more precise.
>
> **2. New Experiments: Addressing Specific Scenarios.** To address your concerns about specific edge cases, we conducted additional evaluations:
>
>   - **Longer Reasoning Chains:** Our test set already covers traces ranging from 30 to ~4,000 tokens. To test extreme lengths, we followed the SafeChain [1] (using a minimum-forcing algorithm) to generate 100 adversarial samples with traces extended to 10,000 tokens. The result below shows that ReasoningShield maintained a high F1 score of 89% on these ultra-long traces, proving its robustness against context dilution in ultra-long reasoning traces.
>
> **Table 3:** Robustness  on Ultra-Long Reasoning Traces
>
> | Model | Accuracy | F1 Score |
> | :---: | :---: | :---: |
> | LlamaGuard-4-12B | 0.495 | 0.281 |
> | **ReasoningShield-3B** | **0.868** | **0.897** |
>
>   - **Domain-Specific Harms:** We evaluated our model on the medical domain safety benchmark MedSafetyBench [2]. We sampled 200 queries and generated corresponding (Query, CoT) pairs. ReasoningShield-3B also achieved a strong F1 score, suggesting that it has learned generalizable risk features that transfer well to specialized domains.
>
> **Table 4:**  Generalization to Medical Domain Harms (MedSafetyBench)
>
> | Model | Precision | Recall | F1 Score |
> | :---: | :---: | :---: | :---: |
> | LlamaGuard-4-12B | 0.861 | 0.413 | 0.559 |
> | **ReasoningShield-3B** | 0.845 | 0.800 | **0.822** |
>
>   - **Multi-turn Interaction:** We followed the design philosophy of mainstream tools like LlamaGuard and WildGuard, focusing on single-turn detection. We argue that multi-turn safety can be effectively managed by decomposing sessions into single-turn checks, which is a standard industry practice.
>
> **3. Qualitative Failure Cases.** We agree that illustrating limitations is vital. Despite strong performance, ReasoningShield is not infallible. In the revised Appendix, we will add a "Limitations & Failure Cases" section.
>
>   - **Example Failure (Knowledge Boundary):** In a query requesting "humorous symptom comparisons," the LRM likened patients to characters like "Shrek" (implying low intelligence) or the "Wicked Witch" (associating patients with "cackling" for sore throats). ReasoningShield misclassified this trace as safe, identifying only the "creative intent" while missing the demeaning undertones.
>
> - **Analysis and Future Work:** This False Negative may stem from a limitation in world knowledge. The model likely lacked the specific cultural context to recognize that associating patients with these specific character traits is offensive rather than merely funny. This highlights a critical direction for future work: integrating RAG or continuous fine-tuning to keep the safety model's knowledge base current and context-aware.
>
> **References:**
>
> [1] Jiang et al., "SafeChain: Safety of language models with long chain-of-thought reasoning capabilities," ACL (2025).
>
> [2] Han et al., "Medsafetybench: Evaluating and improving the medical safety of large language models." NeurIPS (2024).
>
> ---
>
> We sincerely thank you again for your constructive feedback. We hope that these clarifications and additional experiments have fully addressed your concerns. We will incorporate the suggested corrections and key experimental results to further strengthen the manuscript. Should you have any further questions, we would be happy to address them.

---

### Official Review · Reviewer_jj5e · 2025-10-31

**Soundness:** 3
**Presentation:** 4
**Contribution:** 3
**Rating:** 6
**Confidence:** 3

**Summary:**

The paper targets the safety gap in LRMs, where harmful content may appear inside reasoning traces even if final answers look benign. It formalizes CoT moderation, builds a multi-level taxonomy, releases a dataset, and proposes ReasoningShield, a lightweight moderation model trained with a two-stage SFT→DPO pipeline. Experiments report SOTA F1 on CoT moderation and strong OOD generalization, outperforming LlamaGuard-4 and GPT-4o while remaining efficient enough for resource-constrained deployment.

**Strengths:**

1. The paper convincingly motivates why answer-only moderation fails, and why stepwise CoT analysis is required; several mainstream LRMs are shown to degrade on CoT moderation vs. answer moderation.

2.  ReasoningShield-3B reaches ≈91.8% F1, beating LlamaGuard-4 by ~35% and GPT-4o by ~15%; a 1B variant also performs competitively. The two-stage training on small backbones is easy to replicate and practical for deployment. This efficient and accurate detection tool has brought great application value to fields such as online content moderation and reinforcement learning.

**Weaknesses:**

1. Although the dataset mixes sources and shows OOD gains, more naturally occurring product logs or red-teaming traces (with appropriate approvals) would better validate real-world robustness beyond curated constructs.

2. Results are strong aggregate-wise, but more category-wise error analysis would guide deployment boundaries.

3. While the paper demonstrates significant practical value, it lacks theoretical analysis and higher-level methodologies, making it difficult to further generalize the experiences presented.

**Questions:**

1. The paper focuses on reasoning-trace safety, how well does ReasoningShield generalize beyond CoT text to agent reasoning graphs, planning traces, or tool-use trajectories?

2. Reasoning traces often evolve from safe to unsafe or vice versa. Has the model been tested for drift detection—i.e., identifying when the reasoning starts deviating into unsafe territory?


3. With methods like RLVR becoming mainstream, can ReasoningShield act as a reward model within reinforcement alignment loops? For example, could its detection scores be integrated into PPO/RLHF-style training for online safety feedback, effectively turning moderation into an inner-loop verifier rather than a post-hoc classifier?

4. Can ReasoningShield expose a category/level-specific threshold to control over-flagging vs. misses (precision–recall curves per class; safe/potential/harmful)? This would help match heterogeneous org policies and risk tolerances.

---

> ### Author Response · Authors · 2025-11-21
> **Rebuttal to Reviewer jj5e (1/3)**
>
> We thank you for your review and positive feedback.
>
> > **W1:** Suggestions to validate real-world robustness using naturally occurring product logs (with appropriate approvals) or red-teaming traces.
>
> We agree that robustness against red-teaming is critical. While proprietary logs are constrained by privacy, we address this via high-quality proxies (adversarial benchmarks) and a new simulation:
>
> **1. Existing Evidence: Robustness against Adversarial Prompts.** As detailed in **Appendix F.2**, our test set explicitly integrates data from dedicated adversarial benchmarks, as SALAD-Bench and Jailbreak-Bench contain "attack-enhanced" queries designed to bypass safety filters. Our strong performance on these datasets is direct evidence of robustness. For instance, on the Jailbreak-Bench(OSS subset), ReasoningShield-3B achieves an F1 score of 92.3%, whereas LlamaGuard-4 achieves only 69.2%. This demonstrates that our method is highly robust against attackers who intentionally obfuscate prompts.
>
> **2. New Experiment: Robustness against Obfuscated Reasoning Traces.** We interpret your specific concern as referring to attacks where the Reasoning Trace itself is manipulated to evade detection, rather than just the user prompt. To address this, we conducted a new Red Teaming experiment. Inspired by adversarial methods like ReNeLLM [1] and PAIR [2], we selected 100 risky (Query, CoT) samples and applied five distinct obfuscation strategies to the reasoning traces:
>
> - Sensitive Word Substitution and Fuzzification: Replacing key risk terms with synonyms.
> - Sentence Restructuring: Altering word order while maintaining semantic consistency.
> - Noise Insertion: Inserting meaningless characters to disrupt tokenization.
> - Partial Translation: Mixing languages within the trace.
> - Style Transfer: Paraphrasing harmful reasoning into benign-sounding styles.
>
> This generated 500 red-teamed adversarial reasoning traces. The results are shown below:
>
> **Table 1:** Robustness against Red-Teamed Obfuscated Traces
>
> | Model | Accuracy | F1 Score |
> | :---: | :---: | :---: |
> | LlamaGuard-4-12B | 0.333 | 0.500 |
> | **ReasoningShield-3B** | **0.873** | **0.932** |
>
> **Results:** As shown, ReasoningShield maintains high robustness against obfuscated traces compared to the LlamaGuard-4, validating its potential effectiveness against intentional attacks in deployment.
>
> **3. Future Validation:** We acknowledge that "wild logs" represent the ultimate test. By open-sourcing ReasoningShield and our datasets, we provide the necessary infrastructure for the broader research community and enterprise practitioners to validate and extend these findings on diverse, real-world interaction logs in future work.
>
> **References:**
>
> [1] Ding et al., "A wolf in sheep’s clothing: Generalized nested jailbreak prompts can fool large language models easily." NAACL (2024).
>
> [2] Chao et al., "Jailbreaking black box large language models in twenty queries." SaTML (2025).
>
> ---
>
> > **W3:** Concerns regarding the lack of theoretical analysis to generalize the findings.
>
> While our work is empirically driven, it is grounded in the advantage of process supervision over outcome supervision, which we formalized in Section 3.1.
>
> - **Sequential Dependency:** We formulated the safety of the reasoning trace as a sequential dependency problem, where the probability of the safe trajectory depends on every conditional step (Eq. 1): $P(y_{CoT}|Q) = \prod P(t_i | t_{<i}, Q)$. This formulation provides the theoretical basis for why stepwise analysis is necessary: risks can propagate through individual steps ($t_i$).
>
> - **Mitigating the Temporal Credit Assignment Problem:** From a learning theory perspective, traditional "outcome supervision" suffers from severe signal dilution. In long traces, the supervision signal from the final answer becomes diluted when attributed back to specific intermediate steps. A subtle, hidden harmful step effectively acts as "noise" that fails to trigger a negative signal if the final answer is safe. ReasoningShield addresses this by utilizing dense, stepwise supervision, shortening the credit assignment path and preventing risk signals from vanishing over long contexts.
>
> - **Preventing Shortcut Learning:** Optimization objectives using only a binary label are prone to Shortcut Learning.  Models learn to exploit the easy path—specifically that strong "refusal keywords" (e.g., "I cannot")  in the final answer guarantee a "Safe" label. This creates a masking effect, causing the attention mechanism to under-weigh the semantic content of the intermediate, risky steps. By enforcing a stepwise analysis structure, ReasoningShield breaks this shortcut, forcing the model to allocate attention to the process of risk evolution rather than just the final outcome state.
>
> We position this work as a data-centric framework that enables this dense supervision, moving CoT safety from sparse, ambiguous outcome labels to precise, process-oriented oversight.

---

> ### Author Response · Authors · 2025-11-21
> **Rebuttal to Reviewer jj5e (2/3)**
>
> > **W2:** "Results are strong aggregate-wise, but more category-wise error analysis would guide deployment boundaries."
>
> Thanks for this suggestion. We have conducted a breakdown analysis from two perspectives: Risk Categories and Safety Levels.
>
> **1. Analysis by Risk Category**
> - **High Performance (Concrete Risks):** Objective categories like "Economic Harm" and "Prohibited Items" achieve best performance (F1 > 0.93). The specific terminology in these traces makes them easier to classify.
> - **Context-Dependent (The "Safety-First" Behavior):** Subjective categories like "Sex" show High Recall (>93%) but lower Precision (~80%).
>     - **Explanation:** This phenomenon is a direct result of the **"Strict Mode"** configuration used in our main experiments, conservatively flagging "Potentially Harmful" borderline cases as Unsafe to minimize misses.
>     - **Guidance:** While this guarantees that almost all true risks are caught, it leads to some over-flagging. For deployment, organizations can adjust this behavior using the **"Permissive Mode"** to improve Precision based on their specific risk tolerance. (We provide a detailed definition of these modes and their specific trade-offs in our Response to Q4.)
>
> **Table 2:** Performance of ReasoningShield-3B Breakdown by Risk Categories (F1 Score)
>
> | Metric | Violence | Sex | Child | Cyber | Hate | Deception | Political | Rights | Prohib. | Economy |
> | :---: | :---: | :---: | :---: | :---: | :---: | :---: | :---: | :---: | :---: | :---: |
> | **Precision** | 0.814 | 0.787 | 0.901 | 0.906 | 0.815 | 0.863 | 0.913 | 0.899 | 0.937 | 0.944 |
> | **Recall** | 0.960 | 0.937 | 0.877 | 0.928 | 0.907 | 0.910 | 0.875 | 0.947 | 0.929 | 0.927 |
> | **F1 Score** | 0.881 | 0.855 | 0.889 | 0.917 | 0.858 | 0.886 | 0.894 | 0.922 | 0.933 | 0.935 |
>
> **2. Analysis by Safety Level**
>
> - **The "Ambiguity" Challenge:** Errors concentrate in the "Potentially Harmful" class.
>     - **False Negatives:** Occur when the LRM uses heavy "hedging" or protective language (e.g., "theoretically speaking") around risky steps, occasionally leading the model to classify "Potentially Harmful" as Safe.
>     - **False Positives:** Occur when the model interprets a purely educational discussion of a sensitive topic (e.g., "Cybersecurity" concepts) as actionable instructions.
> - **Guidance:** This confirms the necessity of our 3-level taxonomy. Since level 1 detection  is extremely reliable, organizations can confidently automate blocking for clear harmful content, while specifically routing the lower-confidence "Potentially Harmful" subset for manual review to optimize the safety-cost trade-off.
>
> **Table 3:** Performance Breakdown by Safety Level (Accuracy)
>
> | Model | Safe (0) | Potentially Harmful (0.5) | Harmful (1) |
> | :---: | :---: | :---: | :---: |
> | **ReasoningShield-3B** | 0.896 | 0.817 | 0.989 |
>
> > **Q4:** Inquiry regarding category/level-specific thresholds to control the precision-recall trade-off for heterogeneous policies.
>
> Yes, ReasoningShield is inherently designed to support such policy tuning. We implement this via a "Label Mapping Strategy" rather than just numerical thresholds. This design allows users to flexibly adjust risk tolerance without retraining the model, leveraging our 3-level taxonomy.
>
> **1. "Potentially Harmful" as a Policy Knob:** The "Potentially Harmful" level serves as the core mechanism for adjusting risk tolerance to match organizational needs:
>   - **Strict Mode:** Maps "Potentially Harmful" to "Unsafe". This prioritizes high recall, ensuring that even slight or latent risks are flagged. This matches organizations with zero-tolerance policies.
>   - **Permissive Mode:** For scenarios requiring higher creativity or utility, users can map "Potentially Harmful" to "Safe." This prioritizes high precision, flagging only explicit, actionable harm and reducing false positives.
>
> **2. Empirical Validation:** To demonstrate this trade-off, we evaluated both modes on public benchmark. As shown below, switching modes effectively shifts the Precision/Recall balance:
>
> **Table 4:** Performance  Trade-off between Strict and Permissive Modes
>
> | Configuration | Metric | BeaverTails | WildGuard-Test |
> | :---: | :---: | :---: | :---: |
> | **RS-3B (Permissive)**| Precision | **0.908** | **0.964** |
> | **RS-3B (Permissive)**| Recall | 0.659 | 0.720 |
> | **RS-3B (Permissive)** | F1 | 0.764 | 0.824 |
> | **RS-3B (Strict)** | Precision | 0.821 | 0.830 |
> | **RS-3B (Strict)** | Recall | **0.844** | **0.880** |
> | **RS-3B (Strict)** | F1 | **0.832** | **0.854** |
>
> **3. Fine-Grained Probabilistic Control:** Furthermore, since ReasoningShield is an LLM-based model, users can extract the token log-probabilities of the judgment token to draw fine-grained Precision-Recall curves for each category, similar to the mechanism used in LlamaGuard [1].
>
> **Reference:**
>
> [1] Inan et al., "Llama guard: Llm-based input-output safeguard for human-ai conversations." arXiv:2312.06674 (2023).

---

> ### Author Response · Authors · 2025-11-21
> **Rebuttal to Reviewer jj5e (3/3)**
>
> > **Q1:** The paper focuses on reasoning-trace safety, how well does ReasoningShield generalize beyond CoT text to agent reasoning graphs, planning traces, or tool-use trajectories?
>
> This is an exciting direction. While our current work focuses on content safety of textual CoT, the process supervision paradigm utilized in our framework could conceptually generalize to these domains:
>
> - **Planning Traces:** A plan is fundamentally a structured sequence of thoughts. The model's mechanism of "tagging risks in each reasoning step" could be adapted to identify specific unsafe sub-goals or "plan steps" before execution.
>
> - **Tool Use:** Similarly, one may treat tool-use actions (e.g., "Search for bomb recipe") as specific types of reasoning steps. Given that our training data covers procedural knowledge in categories like "Cybersecurity" and "Deception", the model may offer a foundation for detecting harmful intents embedded in procedural steps.  For example, if tool parameters are serialized into text, ReasoningShield could potentially be fine-tuned to detect "malicious tool use."
>
> We acknowledge that agentic structures introduce new complexities beyond the scope of this work, and we consider adapting ReasoningShield for these dynamic environments a highly valuable area for future work.
>
> ---
>
> > **Q2:** Inquiry regarding the model's capability to identify when reasoning deviates into unsafe territory (drift detection).
>
> Yes, our model effectively performs drift detection by design. Unlike holistic classifiers that output a single binary label, ReasoningShield is explicitly trained to execute a stepwise analysis. This granular supervision enables the model to trace the evolution of risk throughout the sequence.
>
> As illustrated in our qualitative examples (e.g.,  Figure 6-8), ReasoningShield decomposes the trace into numbered steps and identifies the transition point. For instance, in Figure 6, the model identifies that while the LRM starts safely with "Framing as Educational," it subsequently drifts into a "Detailed Explanation of Data Sources of Private Information" which constitutes the risk.
> By monitoring this stepwise analysis output, users can pinpoint the exact step $t_i$ where the trace deviates from "Safe" to "Unsafe".
>
> ---
>
> > **Q3:** Assessing the potential of ReasoningShield as a reward model in reinforcement learning loops (e.g., RLVR).
>
> We explicitly highlighted this as a key future direction in Section 6. ReasoningShield is uniquely positioned to serve as a safety reward model in alignment loops for the following reasons:
>
> - **Dense Signal for Process Supervision:** Because ReasoningShield provides a dense, stepwise signal  rather than a sparse binary label, it is ideally suited to function as a process reward model within PPO or RLVR training loops. This enables more granular credit assignment compared to traditional outcome-based rewards.
>
> - **Online Safety Feedback:** Similar to advanced alignment strategies that utilize hybrid rewards from preference and moderation models to optimize for both utility and harmlessness [1-2], our model can provide "online safety feedback" to penalize specific unsafe intermediate steps directly during LRM training. This effectively forces the LRM to "think safely" rather than merely filtering the final output.
>
> This application fundamentally transforms moderation from a post-hoc classifier into an inner-loop verifier, ensuring alignment throughout the entire reasoning trace.
>
> **Reference:**
>
> [1] Yuan et al.,  "From hard refusals to safe-completions: Toward output-centric safety training." arXiv:2508.09224 (2025).
>
> [2] Duan et al., "Oyster-I: Beyond Refusal--Constructive Safety Alignment for Responsible Language Models." arXiv:2509.01909 (2025).
>
> ---
>
> We sincerely thank you again for your constructive feedback. We hope that these clarifications and additional experiments have fully addressed your concerns. We will incorporate the suggested corrections and key experimental results to further strengthen the manuscript. Should you have any further questions, we would be happy to address them.

---

### Official Review · Reviewer_cNvS · 2025-11-01

**Soundness:** 2
**Presentation:** 3
**Contribution:** 3
**Rating:** 4
**Confidence:** 4

**Summary:**

The paper presents ReasoningShield, a framework for moderating the reasoning traces generated by large reasoning models. It highlights that harmful content can appear in the reasoning process even when the final answer looks safe, meaning traditional moderation fails to detect such risks. ReasoningShield formalizes CoT moderation with a structured taxonomy of risk types and safety levels, and introduces a large annotated dataset of reasoning traces. The model uses a two-stage training process combining structured reasoning analysis and preference optimization to improve reliability. Experiments show that ReasoningShield provides more accurate, interpretable, and generalizable moderation of reasoning traces compared to existing moderation systems, demonstrating strong robustness across unseen tasks and models.

**Strengths:**

1. This paper is well-written and clearly presents the dataset description, construction process, and experiments.

2. To improve dataset quality and label reliability, the authors effectively employ methods such as multi-model annotation and human–AI collaboration.

3. Developing a small model that performs competitively in CoT moderation and maintains consistent performance across reasoning traces from various models makes it highly practical and useful.

4. The model demonstrates robust generalization, showing consistent performance not only on the dataset introduced in this paper but also across other benchmarks.

**Weaknesses:**

The dataset and its construction process introduced by the authors are quite impressive; however, there are several issues to point out regarding the experimental setup and analysis of the results.

1. In the pilot study for Table 1, how is the label distribution of the 800-sample dataset? Are the labels well-balanced? Since the F1 score depends on the label distribution, it is important to ensure balance in the dataset.

2. Compared to other models, the F1CoT metric of Claude-Sonnet-3.7 is remarkably low (near zero) in Table 1. Since the authors assume that each model has distinct reasoning traces, they should explain how the reasoning trace of Claude-Sonnet-3.7 differs from others and why it produces such a unique result.

3. While it is encouraging that the small model achieves strong reasoning moderation performance with limited data and simple training, the proposed method itself is not particularly novel.

4. In Table 2, the authors mention that they used Llamaguard’s system prompt for the prompted LLM. However, they should also present results using the ReasoningShield model’s system prompt. As seen in Appendix J.1, ReasoningShield’s system prompt is highly detailed and provides extensive response-generation guidelines, whereas the Llamaguard prompt shown in Appendix J.4 is much simpler. The authors must include this experiment and compare the results.

5. In Table 2, ReasoningShield and Llamaguard perform a three-way classification, and the “potentially harmful” category is merged with “unsafe” for computing the F1 score. In contrast, the prompted LLM performs a purely binary classification task. This difference leads to unequal random baselines: for ReasoningShield and Llamaguard, the random chance of predicting “unsafe” is 2/3 and “safe” is 1/3, whereas for the prompted LLM, both classes have a 1/2 chance.
Since the F1 score considers both precision and recall, this discrepancy introduces unfairness between the two approaches. The authors must provide a justification for this.



+) Typo
L193 WWe->We
L426 cgeneralization->generalization

**Questions:**

Questions are in listed in weakness section.

---

> ### Author Response · Authors · 2025-11-21
> **Rebuttal to Reviewer cNvS (1/3)**
>
> We thank you for your review and insightful feedback.
>
> > **W1 & W2:** Concerns regarding the label balance of the pilot study and the low performance of baselines on Claude-Sonnet-3.7.
>
> 1. **Dataset Distribution and Simulation Design.**  We acknowledge that the label distribution for the pilot study was not explicitly detailed in the initial manuscript. Following the methodology of SafeChain [1], our objective was to simulate a real-world "in-the-wild" scenario. Therefore, we prioritized input consistency by uniformly sampling queries from four safety datasets, allowing the response labels to reflect the models' natural behavior rather than strictly enforcing artificial balance during generation.
> 2. **Explaining Claude-Sonnet-3.7's Performance.** This design directly explains the anomaly observed with Claude-Sonnet-3.7 in Table 1:
>     - **Label Imbalance:** While other LRMs produced a safe-to-unsafe ratio of roughly 5:5 or 6:4, Claude-Sonnet-3.7 was exceptionally well-aligned, resulting in a imbalanced 9:1 ratio.
>     - **Hidden Risks:** The few risk samples it did produce were predominantly "Potentially Harmful". In these traces, risky elements were often deeply interspersed within large volumes of benign text and thereby were more difficult to detect.
>     - **Failure of Existing Tools:** Traditional holistic moderation tools (like LlamaGuard) consistently missed these subtle, "needle-in-a-haystack" risks, leading to the near-zero Recall and $F1_{CoT}$ observed in the original Table 1.
> 3. **New Experiment: Balanced Claude-3.7 Validation.** To confirm that this failure is due to intrinsic model limitations rather than a metric artifact of label imbalance, we conducted a targeted validation study.
>     - **Method:** We prompted Claude-Sonnet-3.7 with all questions from our test set to generate a comprehensive response pool. From this pool, we sampled a specific subset of 200 (Query, CoT) pairs and corresponding (Query, Answer) pairs with a strict **1:1 Safe:Unsafe ratio.**
>     - **Results:** We re-ran the evaluation on this strictly balanced set. The results validate our original conclusion: existing moderation tools struggle significantly with CoT Moderation, confirming that they struggle to parse the complex risk structures in Claude-3.7's reasoning, even when the dataset is strictly balanced.
>
> **Table 1:** Baselines Performance on Balanced  Claude-3.7-Sonnet Dataset
> | Moderation Model | Acc (Ans) | F1 (Ans) | Acc (CoT) | F1 (CoT) | $\Delta$ Acc | $\Delta$ F1 |
> | :---: | :---: | :---: | :---: | :---: | :---: | :---: |
> | LlamaGuard-1-7B | 68.5% | 36.4% | 53.0% | 13.0% | **-15.5%** | **-23.4%** |
> | LlamaGuard-2-8B | 69.5% | 41.9% | 53.0% | 11.3% | **-16.5%** | **-30.6%** |
> | LlamaGuard-3-8B | 66.5% | 29.5% | 53.5% | 13.1% | **-13.0%** | **-16.4%** |
> | LlamaGuard-4-12B | 71.0% | 45.3% | 56.5% | 23.0% | **-14.5%** | **-22.3%** |
> | WildGuard-7B | 73.5% | 53.9% | 56.0% | 21.4% | **-17.5%** | **-32.5%** |
> | OpenAI Moderation | 63.0% | 21.3% | 45.0% | 3.5% | **-18.0%** | **-17.8%** |
>
> **Reference:**
>
> [1] Jiang et al., "SafeChain: Safety of language models with long chain-of-thought reasoning capabilities," ACL (2025).

---

> ### Author Response · Authors · 2025-11-21
> **Rebuttal to Reviewer cNvS (2/3)**
>
> > **W3:** "While it is encouraging that the small model achieves strong reasoning moderation performance with limited data and simple training, the proposed method itself is not particularly novel."
>
> We agree that the individual components of our training pipeline (SFT and DPO) are established techniques. However, we would like to clarify that our core contribution lies not in a new training method, but in a specialized **data-centric framework** to solve the challenging open problem of **CoT Moderation**, which existing moderation models struggle to handle. We did not simply adapt existing data to fit old methods; rather, we identified three unique technical challenges inherent to reasoning traces and designed specific mechanisms to address them:
>
> 1. **Challenge 1: Capturing Fine-Grained Risk Semantics**
>     - **Problem:** Traditional moderation models (e.g., LlamaGuard [1]) rely on binary "safe/unsafe" classification. This limits their ability to detect subtle semantic shifts in the reasoning flow.
>     - **Our Solution:** We introduce a three-level fine-grained taxonomy, explicitly incorporating a "Potentially Harmful" category. This granularity enables the model to capture **nuanced risk semantics** within the sequence. This design is crucial for handling ambiguous traces and facilitates flexible, targeted interventions, thereby significantly boosting practical utility.
>
> 2. **Challenge 2: Localizing Hidden Risks in Sequential Structures**
>     - **Problem:** CoTs are characterized as long-text, multi-step sequences where risks are often sparse or hidden in intermediate steps. Standard "holistic judgment" (treating output as a monolithic black box) fails to perform effective credit assignment, causing models to overlook hidden risks.
>     - **Our Solution:** We design a structured process supervision paradigm for "stepwise, localized analysis." By supervising the model to execute a three-step process: (1) intent analysis, (2) step-level risk tagging, and (3) synthesis into a comprehensive judgment—we force the model to attend to the sequential flow of reasoning. This enables capabilities like "stepwise risk localization" and "enhanced explainability" that existing models lack.
>
> 3. **Challenge 3: Robustness at Decision Boundaries**
>     - **Problem:** Models trained solely on SFT  (e.g., LlamaGuard [1], WildGuard [2], Qwen3Guard [3]) often struggle with hard negative cases.
>     - **Our Solution:** We employ a strategic two-stage training process. We use SFT to establish the structured analysis format on consensus samples, and then apply DPO to optimize performance on "hard negatives" to sharpen the model's judgment on complex boundaries, specifically enhancing robustness.
>
> **While existing moderation models share similar core training methodologies (SFT), we acknowledge that simplicity and efficiency are critical.** Our key contribution lies in identifying and addressing the novel challenges of CoT Moderation, a foundational advance that transcends incremental method adjustments.  We demonstrate that through this specialized, data-centric approach, our lightweight 1B/3B model outperforms massive models like GPT-4o. This "Small-Beats-Large" result highlights the efficiency and the practical value of our work.
>
> **References:**
>
> [1] Inan et al., "Llama guard: Llm-based input-output safeguard for human-ai conversations." arXiv:2312.06674 (2023).
>
> [2] Han et al., "Wildguard: Open one-stop moderation tools for safety risks, jailbreaks, and refusals of llms," NeurIPS (2024).
>
> [3] Zhao et al., "Qwen3Guard Technical Report." arXiv:2510.14276 (2025).

---

> ### Author Response · Authors · 2025-11-21
> **Rebuttal to Reviewer cNvS (3/3)**
>
> > **W4 & W5:** Concerns regarding the impact of prompt design and the fairness of comparing binary (2-way) vs. ternary (3-way) classification tasks.
>
> We address these related concerns together.
>
> 1. **Existing Evidence (Addressing W4).** First, we would like to clarify that we already included this exact ablation study in Table 3 of our original manuscript.
>     - **Original Finding (Table 3):** We demonstrated that for Qwen2.5-72B, replacing the LlamaGuard-3 prompt with our ReasoningShield prompt improved F1 from 79.3% to 88.0%. This confirms the effectiveness of our prompt design and validates your hypothesis that prompt matters.
>     - **Crucial Insight:** However, even with our advanced prompt, the Qwen2.5-72B model still underperformed our ***ReasoningShield-3B***, proving that our methodology contributes significant value beyond prompt engineering.
>
> 2. **New Experiment: Ensuring a Strictly Fair Comparison (Addressing W4 & W5).** To resolve the concern for all baselines and eliminate the "unequal random baseline" issue (W5),  we conducted the experiment suggested in W4: running all prompted LLM baselines using the ReasoningShield prompt (Appendix J.1), forcing them to perform the exact same 3-way classification task as our model.
>
> **Table 2:** Prompted LLM using ReasoningShield prompt on CoT Moderation (F1 Score)
>
> | Model | Prompt Strategy | AIR | SALAD | BeaverTails | Jailbreak | Overall |
> | :---: | :---: | :---: | :---: | :---: | :---: | :---: |
> | **GPT-4o** | w/ LlamaGuard | 0.652 | 0.753 | 0.765 | 0.798 | 0.737 |
> | **Qwen2.5-72B** | w/ LlamaGuard | 0.748 | 0.834 | 0.782 | 0.822 | 0.793 |
> | **Mistral-3.1-24B** | w/ LlamaGuard | 0.600 | 0.760 | 0.707 | 0.731 | 0.700 |
> | **Gemma-3-27B** | w/ LlamaGuard | 0.804 | 0.796 | 0.778 | 0.843 | 0.800 |
> | **GPT-4o** | w/ ReasoningShield | 0.854 | 0.863 | **0.868** | 0.878 | 0.859 |
> | **Qwen2.5-72B** | w/ ReasoningShield | 0.889 | 0.884 | 0.846 | **0.894** | 0.873 |
> | **Mistral-3.1-24B** | w/ ReasoningShield | 0.871 | 0.819 | 0.747 | 0.790 | 0.799 |
> | **Gemma-3-27B** | w/ ReasoningShield | 0.737 | 0.722 3| 0.617 | 0.653 | 0.669 |
> | **ReasoningShield-1B** | w/ ReasoningShield | 0.912 | 0.874 | 0.843 | 0.869 | 0.863 |
> | **ReasoningShield-3B** | w/ ReasoningShield | **0.922** | **0.906** | 0.863 | 0.891 | **0.891** |
>
> **Conclusion:** As shown, even when SOTA models like GPT-4o utilize our structured prompt, their F1 scores remain lower than our lightweight ReasoningShield-3B. This confirms the advantages of our specialized training strategy and high-quality dataset.
>
> 3. **Clarification on Random Baselines (Addressing W5).** Regarding the original experimental setup (Table 2), we clarify that it actually disfavored our model:
>     - **Random Priors:** As you noted, the binary baseline had a 50% chance of predicting "unsafe," while our ternary model had a 66.7% chance.
>     - **Ground Truth:** The actual ground truth of our Test dataset (Figure 3) is approximately 44% Unsafe and 56% Safe.
>     - **Statistical Disadvantage:** The baseline's 50% prior was much closer to the true distribution (~44%) than our model's 66.7% prior. This made the task statistically more difficult for ReasoningShield. The fact that our model still outperformed GPT-4o by ~16% despite this disadvantage further validates its robustness.
>
> ---
>
> > **Typo:** L193 WWe->We L426 cgeneralization->generalization
>
> We will thoroughly proofread the manuscript to correct typographical errors and make it more readable.
>
> ---
>
> We sincerely thank you again for your constructive feedback. We hope that these clarifications and additional experiments have fully addressed your concerns. We will incorporate the suggested corrections and key experimental results to further strengthen the manuscript. Should you have any further questions, we would be happy to address them.

---

### Author Response · Authors · 2025-11-28
**Subject: Notification of Revised Manuscript Upload**

**Dear Reviewers and Area Chair**,

We are writing to kindly inform you that we have uploaded the revised manuscript, which incorporates the additional experiments, clarifications, and textual corrections discussed in our rebuttal responses.

**Key updates include**:

- **Additional Experiments**:

  - **Balanced Pilot Study:** Validated model limitations on a strictly balanced dataset (Section 3.2).

  - **Scaling Behavior:** Analyzed performance across model sizes from 0.5B to 72B (Appendix L.1).

  - **Adversarial Robustness:** Conducted red-teaming on trace obfuscation and ultra-long contexts (Appendix L.2).

  - **Stepwise Baselines:** Compared our end-to-end approach against stepwise application of holistic moderators (Appendix L.3).

  - **Deployment Strategies:** Provided fine-grained error analysis and policy tuning guidance (Strict vs. Permissive Mode) (Appendix L.4).

  - **Stability Analysis:** Reported standard deviations to ensure reproducibility (Appendix L.5).

- **Clarifications**: Refined ambiguous expressions and addressed specific questions on dataset construction.

- **Manuscript Polishing**: We have thoroughly proofread the entire manuscript to correct typographical errors and improve overall readability.

We hope these revisions effectively address your concerns and further strengthen the paper. We remain available to answer any additional questions you may have during the remaining discussion period.

**Best regards,**

**The Authors**

---

### Author Response · Authors · 2025-12-02
**Summary of Rebuttal and Manuscript Revisions**

**Dear Area Chair and Reviewers**,

We are writing to provide a concise summary of our rebuttal process and the revisions made to the manuscript to assist in your final assessment. We have actively addressed all the concrete comments raised by Reviewers cNvS, 9cpv, jj5e and jt2c through additional experiments and explanations.

**1. Consensus on Strengths**

We are encouraged that all the reviewers recognize **the timeliness and the practical value of our work**. Specifically, reviewers highlight the **strong "Small-Beats-Large" performance** (cNvS, jj5e, jt2c), the **high-quality dataset construction** (cNvS, 9cpv), and the **clear presentation** (cNvS, 9cpv, jj5e).

**2. Summary of Responses to Reviewers**

- **To Reviewer cNvS:**

  - **On Pilot Study Balance:** We clarified that the pilot study was to simulate an "in-the-wild" scenario following SafeChain [1]. To rule out label imbalance as a factor for low baseline performance (especially on Claude-3.7), we conducted a new experiment on a **strictly balanced dataset**. Baselines still achieved much lower F1 scores  on CoT moderation, which shows their limitations in detecting hidden risks is an intrinsic model limitation, not a data bias.

  - **On Fairness of Comparisons:** We addressed the concern regarding prompt strategies by running all prompted baselines (e.g., GPT-4o) using our **ReasoningShield system prompt**, forcing them to perform the exact same 3-way classification task. Results confirm that our fine-tuned model still outperforms general-purpose models, validating the efficacy of our training pipeline beyond prompt engineering.

  [1] Jiang et al., "SafeChain: Safety of language models with long chain-of-thought reasoning capabilities," ACL (2025).

- **To Reviewer 9cpv:**

  - **On Novelty&Contribution:** We further revise our contribution statement as "the specialized supervision framework tailored for complex reasoning traces". We clarified how our stepwise analysis paradigm captures risk propagation and subtle semantic evolution compared to holistic "black-box" monitors. We emphasize that our key contribution extends beyond standard training methods (SFT) to the formulation of CoT Moderation and the specialized  framework, which enables our lightweight model (1B/3B) to outperform general-purpose models of a much larger scale (e.g., GPT-4o).

  - **On Scaling & Robustness:** We added scaling laws experiments (validating the efficiency of our 1B/3B scale) and extensive red-teaming (showing resilience against adversarial trace obfuscation and much longer contexts).

  - **On Failure Cases:** We added a qualitative failure analysis (Appendix K.2) discussing knowledge-dependent risks (e.g., cultural metaphors) to illustrate current limitations.

- **To Reviewer jj5e:**

  - **On Real-World Robustness:** We supplemented a new red-teaming experiment using five obfuscation strategies, which shows robustness against adversarial attacks.

  - **On Deployment Guidance:** We provided a fine-grained error analysis by risk category and safety level. We introduced **"Strict" and "Permissive" modes** to help users balance Recall and Precision based on their needs.

  - **On Theoretical Depth:** We enhanced Section 3.1 by incorporating discussions on temporal credit assignment and shortcut learning to ground the advantage of our stepwise supervision paradigm at a theoretical level.

- **To Reviewer jt2c:**

  - **On Additional Baselines:** We implemented the suggested stepwise baseline (applying LlamaGuard to the segmented reasoning steps). Results show that our end-to-end ReasoningShield still achieves better performance (92.5% vs. 85.3%) with significantly lower latency.

  - **On Explainability:** We added a detailed case study in Section 5.2 to demonstrate how our model precisely localizes risks that baselines miss.

  - **On Rigor:** We clarified annotator qualifications and reported standard deviations to ensure reproducibility.

**3. Summary of Manuscript Revisions**

  As highlighted in our previous comment, the revision includes significant enhancements:

  - **Updated Section 3.2:** Revised the pilot study analysis with balanced data findings.

  - **New Section 5.4:** Summarized extended analyses on robustness, scaling, and deployment.

  - **New Appendix L:** Added comprehensive analyses and results for Scaling, Red Teaming, Stepwise Baselines, Deployment Strategies (Strict vs. Permissive Mode), and Stability.

  - **Enhanced Theory:** Added discussion on credit assignment and shortcut learning.

  - **Clarifications:** Refined ambiguous expressions and addressed specific questions.

  - **Global Polishing:** Corrected typos and refined definitions throughout the paper, improving overall readability.

We believe these revisions comprehensively address all the concrete comments from the reviewers and make the paper meet the acceptance standard. Should you have any further questions, please let us know.

**Best regards,**

**The Authors**

---

### Note · Authors · 2026-02-01

I have read and agree with the venue's withdrawal policy on behalf of myself and my co-authors.

---

### Meta-Review · Area_Chair_vRYH · 2026-01-02

**Summary:**

This submission addresses a timely problem: detecting safety risks embedded in chain-of-thought (CoT) reasoning traces of large reasoning models. The paper proposes ReasoningShield, supported by a newly curated dataset, a multi-level risk taxonomy, and a lightweight SFT+DPO training pipeline. However, across reviews, the paper consistently sits at a borderline stage, with concerns around novelty, technical depth, baseline completeness, and experimental sufficiency. While the authors provided an extensive and effortful rebuttal, some major concerns remain below.

1) Novelty and technical contribution. Some reviewers converge on the assessment that the methodological contribution is incremental. While the authors argue that the key novelty lies in “the specialized supervision framework tailored for complex reasoning traces,” this framing remains conceptual.

2) Reinforcement learning integration. One reviewer raised an important question regarding whether ReasoningShield can meaningfully function as a reward or verifier model within RL/RLHF/RLVR-style training loops. While the authors’ response frames this as a promising future direction, this issue is central to the positioning of the proposed approach.

In light of the above considerations, I recommend reject for this submission. With sharper positioning of novelty, stronger baseline design, and concrete exploration of RL-based integration, a future revision could become a strong candidate for acceptance.

**Reviewer Concerns:**

Addressed concerns:
The rebuttal makes a great effort to address several empirical questions raised by the reviewers. In particular, the authors added clarification on dataset construction and annotation quality, included additional analyses on label balance, scaling behavior, and robustness, and incorporated extra baselines such as stepwise application of existing moderation tools.

Outstanding concerns:
Despite these improvements, two core issues remain unresolved: novelty and technical contribution and RL integration.

**Reviewer Scores:**

I think the final scores would still be at the borderline, e.g., 6, 6, 6, 4 or 6, 6, 4, 4.

---

### Decision · Program_Chairs · 2026-01-26

Reject